# Exploring the Effects of Different Drying Methods on Related Differential Metabolites of *Pleurotus citrinopileatus* Singer Based on Untargeted Metabolomics

**DOI:** 10.3390/plants13121594

**Published:** 2024-06-07

**Authors:** Huan Lu, Simin Peng, Ning Xu, Xiaodong Shang, Jianyu Liu, Zhen Xu, Ning Jiang, Haoran Dong, Ruijuan Wang, Hui Dong

**Affiliations:** 1Institute of Edible Fungi, Shanghai Academy of Agricultural Sciences, Shanghai 201403, China; luhuan@saas.sh.cn (H.L.);; 2Institute of Hunan Edible Fungi, Changsha 410013, China; rbtpsm3773@163.com (S.P.);; 3Hunan Provincial Key Laboratory of the Traditional Chinese Medicine Agricultural Biogenomics, Changsha Medical University, Changsha 410219, China; 4Institute of Agro-Food Quality Standard and Testing Technology, Shanghai Academy of Agricultural Sciences, Shanghai 201403, China

**Keywords:** *Pleurotus citrinopileatus* Singer, drying methods, untargeted metabolomic, differential metabolites, metabolic pathway

## Abstract

*Pleurotus citrinopileatus* Singer (PCS) has attracted increasing attention as a raw material for medicine and food. Its quality is greatly affected by the accumulation of metabolites, which varies with the applied drying methods. In this study, we utilize an approach based on ultra-high-performance liquid chromatography/Q Exactive mass spectrometry (UHPLC-QE-MS) to reveal the metabolic profiles of PCS from three different drying methods (natural air-drying, NAD; hot-air-drying, HAD; vacuum freeze-drying, VFD). The results showed that lipids, amino acids and their derivatives were all important secondary metabolites produced during NAD, HAD and VFD treatments, with the key differential metabolites of PCS during drying including fifteen lipids and seven amino acids. Meanwhile, VFD was the best way for long-term preservation of dried PCS. Hot-drying methods, especially HAD, can improve the medicinal component of PCS. Furthermore, KEGG enrichment analysis highlighted 16 pathways and indicated that amino acid metabolism might be the key metabolite pathway for the PCS drying process. Our study elucidates the relationship between drying methods and metabolites or metabolic pathways of PCS to determine the mechanisms affecting the quality of PCS, and finally provides reference values for further development and application in functional food and medications.

## 1. Introduction

*Pleurotus citrinopileatus* Singer (PCS) belongs to the family *Pleurotaceae*, and mainly grows on rotting wood of broad-leaved trees (such as poplar, birch and elm) in the northeast, northwest and southwest of China [1]. It is an important edible fungus, which has potential to be used as a medicinal and functional food. According to the theory of traditional Chinese medicine, PCS is sweet in taste, neutral in nature, beneficial to the lungs and spleen, replenishes the essence and strengthens bones and muscles to treat dysentery. The whole of PCS is commonly used as a herbal decoction or wine to enhance the therapeutic efficacy [2].

Fungal metabolites have historically been categorized into primary, that is, metabolites essential for growth and reproduction, and secondary, which are considered to be rich sources of potential drug candidates and other useful substances [3,4]. According to the literature, the fruit body of PCS is rich in metabolites such as polysaccharides, phenols, flavonoids, terpenoids and alkaloids [5,6], with potential pharmacological activity against neoplastic [3], displays antioxidant activities [3], immune regulatory effects [7], and it has a key role in regulating blood sugar [8] and lowering blood pressure and cholesterol [9]. Meanwhile, the lectin extract from the PCS fruit body exhibits potent anti-inflammatory effects [10]. Not only that, novel polysaccharides found in the PCS mycelium also possess favorable hypoglycemic activity [11] and immunomodulation activity [12]. PCS has attracted widespread attention and is favored by an increasing number of people due to its high nutritious and medicinal value.

As we know, fresh edible fungi typically contain higher moisture content (approximately 85–90%), which is not conducive to long-term preservation. Proper post-harvest processing not only contributes to reducing post-harvest losses, but mushrooms can also be preserved for downstream processing to enhance economic value. Among the many method options, drying remains the most common method for the long-term preservation of mushrooms [13]; meanwhile, it is also the first key step for the quality formation of medicinal materials [14]. In the early stage of drying, PCS is still a living organism, which focuses on breathing. Under the influence of environmental parameters, especially the temperature [15], the quality of PCS changes constantly during the process of drying. It was reported that drying methods can significantly affect chemical components and even promote the formation of some new chemical components. So, the quality of PCS is closely related to its chemical components, the composition of which also varies with drying methods. Currently, numerous researchers pay attention to this problem. For example, Ma Y et al. (2020) studied the effect of various drying methods on the lipid profiles of truffles. Results demonstrated that freeze-drying had little effect on the lipid composition of truffles so as to be a better way than heat-drying for truffle processing [16]. Kibar et al. (2021) determined the effects of six different drying methods on the postharvest quality and nutritional properties (pH value, ash, protein, color, total phenolic content, total flavonoid content, antioxidant activity and element contents) of *P. ostreatus* mushroom, and showed that natural drying was the most effective treatment among drying methods, and it can be used successfully for long-term storage of this mushroom [17]. The researchers think that VFD is a good choice for drying Schizophyllum commune, due to the fact that this method could better maintain the primary color and natural form. Meanwhile, VFD samples showed superior retention rates for total organic acid and umami concentration values [18]. As described above, most current research focuses on the impact of drying methods on mushroom physical characteristics, non-volatile taste components, and nutrient retention, resulting in a lack of information concerning the influence of the drying processing method on the DMs of mushroom. In this paper, we propose that by elucidating the relationship between drying methods and metabolites or metabolic pathways of PCS to determine the mechanisms affecting the quality of PCS we can inform resource utilization.

In this study, PCS was selected as the research object, and a UHPLC-QE-MS metabolomics approach was utilized to analyze the types and relative contents of metabolites in PCS from three different drying methods (NAD, HAD and VFD). Based on the above data, PCA, OPLS-DA and HAC were used to explore the influence of drying methods on whole metabolic profiles of PCS. Volcano maps, Venn diagrams, and metabolic pathway analysis were applied to clarify the DMs and reveal the effect of drying treatments on metabolic substances and the mechanism of DM conversion in PCS. The aims of this study were as follows: (i) Explore the DMs of PCS under different drying treatments and to screen potential marker DMs for the identification of different drying methods. (ii) Offer some new insights into the different drying processes and provides a major contribution to research on the main bioactive compounds that affect the quality of PCS.

## 2. Materials and Methods

### 2.1. Fresh Mushroom and Reagents

Fruiting bodies of the golden oyster mushroom (*Pleurotus citrinopileatus* Singer, PCS), which is widely grown and planted in China, were harvested from the Institute of Edible Fungi, Shanghai Academy of Agricultural Sciences on 10 April 2022. And mushrooms were identified as *Pleurotus citrinopileatus* Singer by ITS sequencing (Sangon Biotech (Shanghai) Co., Ltd., Shanghai, China).

Methanol (LC/MS grade) and acetonitrile (LC/MS grade) were purchased from CNW (CNW Technologies, Düsseldorf, Germany), ammonium acetate (LC/MS grade) was purchased from Sigma Aldrich (Sigma Aldrich, St. Louis, MO, USA), ethanoic acid (LC/MS grade) was purchased from Fisher Scientific (Hampton, NH, USA) and Ultrapure water was obtained with the Milli-Q water system purchased from Millipore (Bedford, MA, USA).

### 2.2. Mushroom Preparation

The detailed process was as follows (Figure 1):(1)Material selection: When the fruit body color of PCS changed to fresh yellow, and this can also be golden yellow, the fruiting bodies free from blemishes and mechanical damage were collected. And mushrooms of uniform size (including the cap and stipe) and maturity were selected as experimental material.(2)Mushroom picking: The root of PCS was carefully pulled and picked out. Or a knife was used to cut the roots off at slightly above where the root began.(3)Washing: The picked fruiting bodies were cleaned with running water to remove remaining attached soil, and then the moisture on the surface was wiped with filter papers.

### 2.3. Drying Process

Natural air-drying (NAD): 1000 g fresh PCS was placed into an air-drying tray, and the samples were then placed in a well-ventilated location with abundant sunlight. The average temperature was 40 ± 2 °C and the relative humidity was 70 ± 10%. The drying time lasted for 2 to 3 days until the final moisture content was controlled at approximately 12%.

Hot-air-drying (HAD): 1000 g fresh PCS was dried in a forced air circulation oven (GZX-9240MBE, Boxun Equipment Ltd., Shanghai, China) at 55 °C for 2 to 3 h. Samples were dried until the final moisture content was controlled at approximately 12%.

Vacuum freeze-drying (VFD): 1000 g fresh PCS was dried by a freeze drier (FD-5 LD plus, SIM International Group Co., Ltd., San Jose, CA, USA). The vacuum degree was 20–40 Pa. Cold trap temperature was −50 °C. The drying chamber was −40 °C. Drying was carried out for 24 h until the moisture content was controlled at approximately 12%.

All dried PCS from the three drying methods (NAD, HAD and VFD) were ground into a powder (passed through a 425 μm sieve), and then stored at −80 °C until analysis.

### 2.4. Untargeted Metabolomics Analysis by UHPLC-QE-MS

#### 2.4.1. Sample Preparation and Extraction

Sample preparation for untargeted metabolomics was performed according to the manufacturer’s instructions. Three replicates per group were set. Briefly, a 20 mg sample was weighted to an EP tube, and 1000 μL extract solution (methanol–water = 3:1, with isotopically labeled internal standard mixture) was added. Then, the mixture was homogenized at 35 Hz for 4 min and sonicated for 5 min in an ice-water bath. The homogenization and sonication cycle were repeated 3 times, followed by standing at −40 °C for 1 h and centrifuged at 12,000 rpm (RCF = 13,800× *g*, R = 8.6 cm) for 15 min at 4 °C. The resulting supernatant was transferred to a fresh glass vial for analysis. The quality control (QC) sample was prepared by mixing an equal aliquot of the supernatants from NAD, HAD and VFD PCS.

#### 2.4.2. LC-MS/MS Conditions

The untargeted metabolomics analysis was conducted according to reported methods previously described by Zhang et al. [19] with minor modifications. Briefly, LC-MS/MS analyses were performed using an UHPLC system (Vanquish, Thermo Fisher Scientific, Waltham, MA, USA) with a UPLC BEH Amide column (2.1 mm × 100 mm, 1.7 μm) coupled to a Q Exactive HFX mass spectrometer (Orbitrap MS, Thermo). The mobile phase consisted of 5 mmol/L ammonium acetate and 5 mmol/L acetic acid in water (pH = 9.75) (A) and acetonitrile (B). The elution program was as follows: 0–0.7min, 1% acetonitrile elution, flow 0.35 mL/min; 0.7–9.5 min, 1–99% acetonitrile elution, flow 0.35 mL/min; 9.5–11.8 min, 99% acetonitrile elution, flow 0.35–0.5 mL/min; 11.8–12 min, 99–1% acetonitrile elution, flow 0.5 mL/min; 12–14.6 min, 1% acetonitrile elution, flow 0.5 mL/min; 14.6–14.8 min, 1% acetonitrile elution, flow 0.5–0.35 mL/min; 14.8–15 min, 1% acetonitrile elution, flow 0.35 mL/min. Auto-sampler temperature was 4 °C; column temperature, 35 °C; and the injection volume was 2 μL.

The QE HFX mass spectrometer was used for its ability to acquire MS/MS spectra in information-dependent acquisition (IDA) mode in the control of the acquisition software (Xcalibur, version 4.5, Thermo). In this mode, the acquisition software continuously evaluates the full scan MS spectrum. The ESI source conditions were set as following: sheath gas flow rate as 30 Arb, Aux gas flow rate as 10 Arb, capillary temperature at 350 °C, full MS resolution as 120,000, MS/MS resolution as 7500, collision energy as 10/30/60 in NCE mode, spray voltage as 4.0 kV (positive) or −3.8 kV (negative), respectively.

The raw data were converted to the mzXML format using ProteoWizard version 3 and processed with an in-house program, which was developed using R v4.0.0 and based on XCMS, for peak detection, extraction, alignment, and integration. Then, an in-house MS2 database [20] was applied in metabolite annotation. The cutoff for annotation was set at 0.3.

#### 2.4.3. Data Processing and Multivariate Analysis

Multivariate statistical analysis has the advantage of simplifying and reducing the dimension of high-dimensional and complex data on the premise of retaining a large amount of original data; this makes it more popular for analyzing and researching the increasingly large and complex datasets of metabolomics. In this study, principal component analysis (PCA), an unsupervised analysis method, was carried out on all samples, including quality control samples (QC), to elucidate the total metabolic differences and the variation degree among the samples. Orthogonal projections to latent structures discriminant analysis (OPLS-DA), a more supervised machine learning data analytical method, was applied to figure out values of variable importance in projection (VIP) of each metabolite to screen out DMs in the pairwise comparisons. The accumulation pattern of metabolites among different PCS was analyzed by hierarchical cluster analysis (HCA). DMs were annotated and classified using the Kyoto Encyclopedia of Genes and Genomes (KEGG) database (http://www.kegg.jp/kegg/pathway.html), accessed on 15 May 2023.

### 2.5. Statistical Analysis

All PCS samples had three full replicates including the drying procedure, and the data were expressed as mean ± standard deviation (SD). In order to minimize the impact of systematic errors on the results and to better highlight the biological significance of the results, the raw data were preliminarily managed by following four steps: filtering deviation value, filtering missing value, filling missing value and normalizing data [21]. Then, the processed data were analyzed by SIMCA 14.1 software for PCA OPLS-DA and HCA multivariate statistical analysis. The *t*-test was applied using SPSS version 17.0 software, and the level of *p* < 0.05 was considered statistically significant.

## 3. Results

### 3.1. Overview of the Metabolic Profiles of PCS Collected from Three Different Drying Methods

In order to evaluate the effect of the different postharvest processing on PCS metabolome, the samples subjected to three drying methods, namely, NDA, HAD and VFD, were used for metabolic profiling based on the untargeted metabolomics approach. The total ion chromatogram (TIC) diagram of the quality control (QC) samples and the extract ion chromatogram (EIC) plots of internal standard in the QC sample are shown in Appendix A, and the results showed that the retention time and response intensity of internal standard in QC samples had good consistency, indicating good stability of the data acquisition of mass spectrometers in this study.

Based on the local metabolite database, qualitative and quantitative MS analyses were conducted on the metabolites in the samples. In total, 885 metabolites were identified (Appendix A) and divided into 13 categories (Appendix A). In the pie chart, four superclasses including (1) 87 organoheterocyclic compounds, (2) 54 organic oxygen compounds, (3) 11 organic nitrogen compounds and (4) 3 organosulfur compounds were merged to one group named organic compounds. Additionally, eight other superclasses were (5) 221 lipids and lipid-like molecules, (6) 110 amino acids and peptides and analogues, (7) 62 flavonoids and phenolic acids, (8) 57 alkaloids and derivatives, (9) 39 terpenoids, (10) 31 benzenoids, (11) 27 nucleotides and derivatives, (12) 21 organic acids and derivatives and (13) 161 unclassified metabolites.

### 3.2. Visualize Analysis of Three Drying Methods of PCS

To assess the differences in the metabolic profiles of PCS between the three drying methods (NAD, HAD and VFD), an unsupervised PCA approach was used to conduct a comprehensive analysis of different PCS samples, including QC samples. In the PCA 3D plot (Figure 2A), QC samples (orange dots in the Figure) are closely clustered, indicating good analytical repeatability and stability of the experiment. Furthermore, in the comparison among NAD, HAD, VFD and QC groups, the first two principal components explained 56.91% of the change, with 32.67% for PC1 and 24.24% for PC2. And a clear discrimination can be observed corresponding to the drying method in the PCA 3D score plot, indicating that the drying methods can affect the metabolite composition in PCS. This conclusion was also verified by the hierarchical heatmap which showed that all PCS samples from the same drying method were clustered together, denoting the reliability of the metabolic profiling data (Figure 2E).

In order to better visualize the differences between the samples, samples were compared in HAD and NAD, VFD and HAD and VFD and NAD. Meanwhile, specific differences of PCS samples were then explored using a supervised orthogonal projection to latent structures discriminant analysis (OPLS-DA) model. As shown in Figure 2B–D, in PCA plots, the samples in different groups are situated in different quadrants on the left and right sides of the *y*-axis. Samples of the same group are clustered together, whereas samples of different groups are clearly separated. Also, OPLS-DA analysis revealed similar results to PCA analysis. In order to avoid over-fitting of the model, the OPLS-DA model was cross-validated with 200 permutation tests. For HAD and NAD, R2Y(cum) = 0.99, Q2(cum) = 0.963; for VFD and HAD, R2Y(cum) = 0.98, Q2(cum) = 0.954; for VFD and NAD, R2Y(cum) = 0.98, Q2(cum) = 0.972. These results indicate that the OPLS-DA model was stable and highly reliable for this dataset and support the validity of the model in further screening different metabolites.

### 3.3. Differential Metabolite (DM) Analysis of PCS in Three Different Drying Methods

#### 3.3.1. DMs Profiles in Different Drying Treatments of PCS

To identify the differentially expressed metabolites among the three drying methods of PCS, the variable importance in projection (VIP) score of OPLS-DA model as a univariate analysis was applied to rank the metabolites that best distinguished the different groups in this study. Notably, the VIP value reflects the influence of every metabolite ion on classification, and VIP > 1 indicates a significant contribution to the separation of a sample group. In addition, the fold change value (FC) for each differential metabolite was transformed as Log2, and the corresponding *t*-test *p*-value was transformed as −Log10. They were used as a univariate approach for screening differentially abundant metabolites. Finally, metabolites with VIP > 1, *p*-value < 0.05 and FC > 2 or FC < 0.5 were selected [22,23] and regarded as significant differential metabolites (DMs) between PCS samples. Overall, the contents of approximately half of the metabolome (480 metabolites) were significantly altered, showing that the choice of drying approach has a great influence on the metabolite content in PCS. Among them, there were 265 significant DMs between HAD and NAD (85 up-regulated and 180 down-regulated), 190 DMs between VFD and HAD (156 up-regulated and 34 down-regulated) and 276 DMs between VFD and NAD (144 up-regulated and 132 down-regulated) (Appendix A; Figure 3A).

The DMs produced in PCS from different drying methods were further classified and compared. According to Figure 3B, DMs can be divided into 12 different categories (not including the other category), mainly concentrating on (1) alkaloids and derivatives, (2) amino acids, peptides and analogues, (3) lipids and lipid-like molecules, (4) organoheterocyclic compounds and (5) flavonoids and phenolic acids. It was shown in pairwise comparisons (HAD vs. NAD and VFD vs. NAD), that more up-regulated lipids and lipid-like molecules were detected in the NAD group than the other two groups. And the lipids and lipid-like molecules were mainly (10E,12Z)-9-HODE, (9xi,10xi,12xi)-9,10-Dihydroxy-12-octadecenoic acid, 15-Methylpalmitate, 9,10-Epoxyoctadecenoic acid and 2-Hydroxystearic acid (Appendix A). On the other hand, for alkaloids and derivatives, amino acids, peptides and analogues, organoheterocyclic compounds and flavonoids and phenolic acids, these DMs had higher up-regulated numbers in the HAD and VFD groups than the corresponding NAD group. These results indicated that some physiological and metabolic activities might be activated or inhibited by temperature. Furthermore, when comparing VFD with HAD (column charts with vertical stripes), it was found that except for organosulfur compounds, the number of up-regulated metabolites detected in the PCS of HAD in other categories was significantly higher than the corresponding number of down-regulated metabolites. The result is consistent with the volcano diagram result, which indicated that HAD enhanced the contents of DMs in PCS.

#### 3.3.2. Effects of Different Drying Methods on the DMs of PCS

To gain insight into how these metabolites behave in PCS treated by different drying methods, the compound abundance of each drying method was calculated and is displayed as bar graphs in Figure 4 for comparative analysis. The results suggested that the amino acids as well as the benzenoids, organoheterocyclic compounds, organosulfur compounds and nucleosides had the higher content levels under HAD and VFD treatment conditions than in the NAD treatment. Furthermore, the flavonoids and phenolic acids, organic oxygen compounds and terpenoids had the maximum contents by the NAD method. In addition, lipids exhibited the lowest abundance under the VFD treatment.

The identification of key metabolites might help to analyze the effects of different drying methods on PCS metabolites. An upset diagram (Figure 5A) was created to show the unique and common metabolites that were differentially expressed between the HAD vs. NAD group, VFD vs. NAD group and VFD vs. HAD group. As shown in this diagram, 107, 72 and 21 unique differential metabolites were identified in the HAD vs. NAD group, VFD vs. NAD group and VFD vs. HAD group, respectively. Meanwhile, this diagram depicts the shared 58 DMs which are considered as key metabolites among NAD vs. HAD vs. VFD. In order to better understand the 58 DMs’ distribution, the classification of the 58 DMs (Appendix A) are shown in Figure 5B. Among them, the proportion of key metabolites associated with the lipids and lipid-like molecules was the largest (25.86%), followed by amino acids, peptides and analogues (12.07%).

To comprehensively analyze the changes in key differential DMs caused by different drying methods, heatmaps and box-plots of the DMs were obtained by classification. For lipids and lipid-like molecules (Figure 6), their content levels in the NAD and HAD samples were significantly higher than those in the VFD group, except dioscoretine and PC (20:5(5Z,8Z,11Z,14Z,17Z)/20:3(5Z,8Z,11Z)). Amino acids, peptides and analogues (Appendix A) were also compared, showing similar trends. L-lysopine, leucyl-threonine, glutaminylphenylalanine and tyrosyl-gamma-glutamate in the HAD group and tyrosyl-hydroxyproline and gamma-l-glutamyl-l-pipecolic acid in the NAD group exhibited significantly higher content levels than the corresponding compounds in the VFD group. Regarding other DMs (Appendix A), we found that their content levels are very high in the HAD group. And for 2-aminonaphthalene, tryptamine, 5′-methylthioadenosine, biotin amide and betaine, HAD and VFD resulted in high content levels, which meant the HAD and VFD methods could better retain their contents compared to the NAD method. These results indicate that the drying method significantly affected the composition of the pretreated PCS, and this difference may be caused by the difference in treatment temperature.

### 3.4. Metabolic Pathway Analysis

To further explore the biological mechanisms underlying these DMs in PCS, KEGG pathway enrichment analysis was conducted. Results show that a total of 107 pathways were involved in the KEGG enrichment analysis. Among them, 265 DMs between NAD and HAD were distributed in 88 pathways, 276 between NAD and VFD were distributed in 99 pathways, and 190 between HAD and VFD were distributed in 70 pathways (Appendix A). After statistical analysis, it was found that 57 metabolic pathways were involved in three groups, and most of the DMs were assigned to the metabolic pathways, secondary metabolite biosynthesis and microbial metabolism in diverse environments; the bubble charts show the top 20 KEGG enriched metabolic pathways (Figure 7). In particular, 16 metabolic pathways were identified with *p*-values less than 0.05 (2, NAD vs. HAD; 11, NAD vs. VFD; and 3, HAD vs. VFD). These pathways mainly concentrated in amino acid metabolism, including “phenylalanine metabolism”, “tryptophan metabolism”, “arginine biosynthesis”, “arginine and proline metabolism” and “cyanoamino acid metabolism”. In addition, they are also involved in “vitamin B6 metabolism”, “ubiquinone and other terpenoid-quinone biosynthesis” “monobactam biosynthesis” “aminoacyl-tRNA biosynthesis”, “ABC transporters” and the “sulfur relay system”.

## 4. Discussion

Sample collection was processed through different drying methods, during which the flavor, color, texture and active substances of PCS were changed. In this study, a UHPLC-QE-MS metabolomics approach was successfully used to detect, in all PCS samples, a total of 885 metabolites obtained from the three drying methods, which were identified and grouped in 13 different categories. It can be seen that the metabolite composition of PCS is complex and diverse. Through pairwise comparison of samples in three groups, it was found that, no matter whether NAD, HAD or VFD is adopted, the main metabolite types of PCS are concentrated in classes such as lipids and amino acids. This finding is similar to the metabolite analyses results of *P. ostreatus* and *P. citrinopileatus* reported by Li et al. [24]. However, further data analysis revealed that the amounts and contents of many metabolites were distinctly different between each group, indicating that drying methods can exert an important effect on metabolite formation.

Changes in amino acid metabolites in response to different drying methods are usually closely related to the complex biochemical reactions of protein in the PCS fruit body. Mushrooms are high in protein, and the content of protein in PCS is generally higher than other mushrooms (typically 17.5% protein), with the content being found from 23% to 44% [5,25]. During drying, proteins are hydrolyzed into small peptides and amino acids by the action of endogenous enzymes and microorganism [26]. Data analysis showed that amino acid-related metabolic pathways were significantly enhanced under the HAD and VFD treatment compared to NAD, and the contents of many metabolites involved in amino acid biosynthesis were also found to increase. Among them, L-Phenylalanine, L-Tyrosine, L-Histidine and L-Valine were the primary components of bitter amino acids, and they were predominant in the HAD and VFD groups. This conclusion can also be supported by the phenomenon observed by Hou [5]. However, it is gratifying that L-Arginine, L-Aspartic acid and L-Proline are the main contributors to umami, and their content is also increasing in the HAD and VFD groups, which can counteract the astringent or bitter taste and increase the umami taste of PCS [27]. Furthermore, ornithine has a flat or sweet taste and masks bitterness, and can be considered as a potential antioxidant food material that can be added to many kinds of food to prevent hepatic injury [28]. Pyroglutamic acid is important for free amino acid transportation, which can enhance the meaty and umami flavor of PCS [29]. All these amino acids are related to the ultimate flavor of PCS. It was observed from Appendix A that the amounts of bitter amino acids were the most abundant in HD, followed by VFD. The levels of umami and sweet amino acids were significantly enhanced (*p* < 0.05) by VFD compared to HAD. These results indicate that VFD is a more suitable method of drying to enhance the flavor value of PCS. In addition, VFD retained amino acid associated with umami better than NAD and HAD, which may be related to non-enzymatic oxidative reactions of amino acids during processing. Nucleotides and their derivatives have been shown to be the main functional components and flavor substances of many edible mushrooms, and the nucleotide content affects the umami of PCS, especially in the HAD process where nucleotides are prone to degradation due to prolonged high-temperature exposure, which affects the quality of the commodity. Yang found that there were significant differences in the taste of *Dendrobium officinale* flower tea prepared by different drying methods due to changes in the amino acid content of glutamine and pyroglutamic acids contained in the samples during the drying process, and the metabolism of such amino acids is closely related to the taste profile [30].

During the drying process, the gradual reduction in relative water content of the PCS causes water deficiency in cells, resulting in drought stress. This abiotic stress is capable of activating complex defense systems in the PCS, including the accumulation of osmolytes and antioxidants [31]. According to the results in our study, the types of up-regulated amino acid metabolites as well as the relative content of amino acids (e.g., proline, alanine, histidine and valine) were significantly increased in the VFD and HAD groups compared to the NAD group. It is well known that amino acids are an important nitrogen source for plants. The accumulation of amino acids in PCS is thought to be the result of nitrogen mobilization fueled by protein degradation that occurs during dehydration [32]. The significant accumulation of amino acids and derivatives regulates plant defense against drought stress by osmotic balancing and maintaining the stability of the cell membrane structure. The accumulating amino acids are transport into the fruit body for storage to fuel growth when PCS is rehydrated. The amino acid metabolism also appears to reflect this process. The pathways linked to amino acid metabolism were enhanced in the HAD and VFD groups.

In our study, some polyphenols levels, such as 3-Hydroxycinnamic acid, Phenylacetaldehyde and 2-Hydroxycinnamic acid, increased in the HAD treatment, which might be due to the fact that polyphenols with a protective function in fungi and their biosynthesis were up-regulated under drought and high-temperature stress, aiming at promoting antioxidant activity and scavenging free radicals [33]. Generally, PCS is still a living organism in the early stage of drying process. And the operating temperature is generally much higher than the temperature suitable for PCS growth under hot-air-drying processes. In this situation, high-temperature stress will be induced. The accumulation of metabolites, especially the polyphenols, in PCS under high-temperature stress conditions changes significantly to facilitate the interaction of biochemicals and molecular processes within the organism to adapt to the stressful environment [34]. 3-Hydroxycinnamic acid and 2-Hydroxycinnamic acid belong to hydroxycinnamic acid derivatives isolated from *Cinnamomum cassia* Presl. They are polyphenols with bioactive properties, including antioxidant and anti-inflammatory properties [35]. As shown in our results, polyphenols involved in the phenylalanine metabolic pathway were found to be differentially expressed. The phenylalanine metabolic pathway is an important secondary metabolic pathway in plants. Phenolics are the abundantly available secondary metabolites derived from phenylalanine via the secondary metabolic pathway. The phenylalanine content in the HAD group increases under high-temperature stress, thereby promoting the production of polyphenols and flavonoids. This is consistent with the findings of Wang et al. (2022), where Arabidopsis subjected to high-temperature stress showed an increase in the expression of flavonoids [36].

We also found that a number of biologically active metabolites were significantly up-regulated in HAD group. Tryptamine is a common functional group of biologically active compounds, such as neurotransmitters and psychedelic drugs, and serves as an intermediate for the biosynthesis of psilocybin. It exists in most plants and mushrooms [37]. 5′-Methylthioadenosine is an important metabolite in the methionine salvage pathway and serves as a precursor for the regeneration of methionine and SAMe [38]. Betaine plays a key role in the antioxidant and proinflammatory system [39], and serving as an osmoprotectant and methyl donor [40]. But another interesting finding contrary to the above conclusion emerged from our results. Hexamethylquercetagetin, a phytochemical flavonoid compound isolated as a new metabolite in citrus plants [41], exhibits powerful antimicrobial activity [42]. It can be seen from Appendix A that the hexamethylquercetagetin content was extremely high in phenolic DMs from the NAD group (around 95.69%), and this was almost three times more than the HAD group and the VFD group, respectively. Although some phenols are sensitive to changes in high temperature, it has a high tolerance to cold stress [43]. This conclusion is inconsistent with our conclusion. The reason for this observation is currently unknown and requires further investigation. On the other hand, phenolic acids were easily decomposed by pyrolysis; the drying process has a large effect on the phenolic acid content of PCS, resulting in changes in the flavor of PCS after drying. The bright color of PCS was due to the rich content of compounds such as anthocyanins and other flavonoids, which have less impact on the stability of anthocyanins during the VFD process, and also inhibit the browning of PCS, so that VFD can maintain the color of PCS.

Lipids are a crucial factor in determining the qualities of mushrooms, such as flavor, palatability and nutritive value [44]. 9,10-Epoxyoctadecenoic acid, a member of the linoleic epoxide (EpOME) class, is synthesized via the conversion of linoleic acid [45]. 9,10-Epoxyoctadecenoic acid is one of the most abundant lipid DMs in PCS dried by NAD, representing 51.74%. Its contents were ranked as follows: NAD>HAD>VFD. Meanwhile, similar content changing patterns were also found in 2-Hydroxystearic acid, 13-OxoODE and (9xi,10xi,12xi)-9,10-Dihydroxy-12-octadecenoic acid. 13-OxoODE and (9xi,10xi,12xi)-9,10-Dihydroxy-12-octadecenoic acid are the oxidized linoleic acid metabolites [46,47], and 2-Hydroxystearic acid is a regio-isomer of Hydroxystearic acid. Hence, the variation in these metabolites showed that compared with other methods, VFD displays advantages which enable the preservation of the fresh profiles of PCS, because it can reduce lipid oxidation. As we know, lipid oxidation is a major cause of quality deterioration in food, and is strongly affected by high-temperature and high-oxygen environments. On one hand, VFD was carried out at a low temperature (−40°C). The low temperature kept cells dormant and made their walls insusceptible to destruction, which might result in fewer lipids being released. On the other hand, VFD reduced the prolonged exposure of PCS to air. Oxygen plays a fundamental role in the drying progress because of its involvement in chemical reactions and biological processes that impact the sensory and quality profile, such as Maillard reactions, Strecker degradation, aldol condensation and lipid oxidation [48]. Notably, acylcarnitines were another group of significant DMs, as important intermediaries of lipid metabolism, and being closely related to energy homeostasis. In our study, the HAD and VFD treatments resulted in increased levels of acylcarnitines including propionylcarnitine, Isobutyryl-L-carnitine and 2-Methylbutyroylcarnitine. According to Ahmed et al.’s (2020) study, acylcarnitines are inhibitory neurotransmitters, regulating neuronal excitability and playing a role in anxiety, stress and emotional regulation [49].

Furthermore, volatile substances are the key factors affecting the flavor of PCS. Volatile carbonyl compounds like aldehydes, ketones, esters and alcohols are the important volatile flavor components of PCS. They are usually stored in bound form so a relevant amount of them can be released due to thermal processes. Meanwhile, the α-dicarbonyl compounds generated during heat treatment or storage have been recognized as a very important precursor of flavor formation [50]. But the non-targeted metabolomics method used in this study was mainly to detect the non-volatile metabolites. The volatile metabolites needed to be detected and identified using other methods, such as GC-MS, GC-IMS, HS-SPME-GC-MS/MS. In the future, the above shortcomings will also be the direction and goal of our continued research.

## 5. Conclusions

The study investigated the metabolic profiles of three PCS samples obtained from NAD, HAD and VFD, respectively. Firstly, a total of 885 metabolites were tentatively identified and 480 DMs were screened, which showed the choice of drying approach has a great influence on the metabolite content in PCS. These DMs of the dry processing of PCS across 13 classes of which lipids, amino acids and their derivatives were significant secondary metabolites. Among them, 15 lipids and seven amino acids were considered to be key DMs for PCS in the response to drying treatments. In congruence, KEGG enrichment analysis highlighted 16 pathways with *p*-values less than 0.05, among which 2 pathways for NAD vs. HAD, 11 pathways for NAD vs. VFD and 3 pathways for HAD vs. VFD, concentrated in amino acid metabolism (including “Phenylalanine metabolism”, “Tryptophan metabolism”, “Arginine biosynthesis”, “Arginine and proline metabolism” and “Cyanoamino acid metabolism”).

The variation in these DMs affects the quality of PCS as a medicinal and food material. By comparing all three drying methods, we found that VFD was the best method (compared to both NAD and HAD) for drying the studied PCS from the perspective of flavor and long-term preservation, because the lipid content of the PCS treated with VFD was lowest, especially the oxidized linoleic acid metabolites like 9,10-Epoxyoctadecenoic acid. It indicated that the VFD can reduce lipid oxidation and enables the preservation of the fresh profiles of PCS. Furthermore, VFD also significantly enhanced the levels of amino acids (especially umami and sweet). From a medicinal chemistry perspective of PCS, hot-drying is better than cold-drying. Compared with NAD, HAD can be used to effectively control for length of time and reduce the prolonged exposure to hot air, and allow better retention of bioactive compounds.

Collectively, the present study suggests the huge potential of VFD in the field of PCS preservation and offers an important reference for HAD and NAD to producing bioactive metabolites from PCS; this can provide a solid theoretical reference for development and industrial application.

## Figures and Tables

**Figure 1 plants-13-01594-f001:**
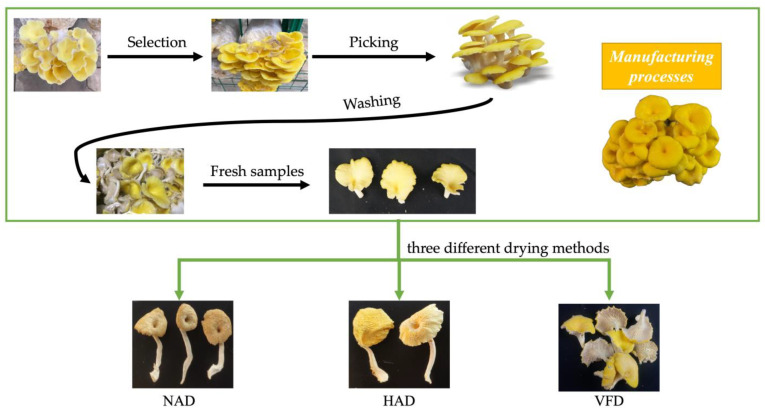
Manufacturing processes of PCS through three different drying methods.

**Figure 2 plants-13-01594-f002:**
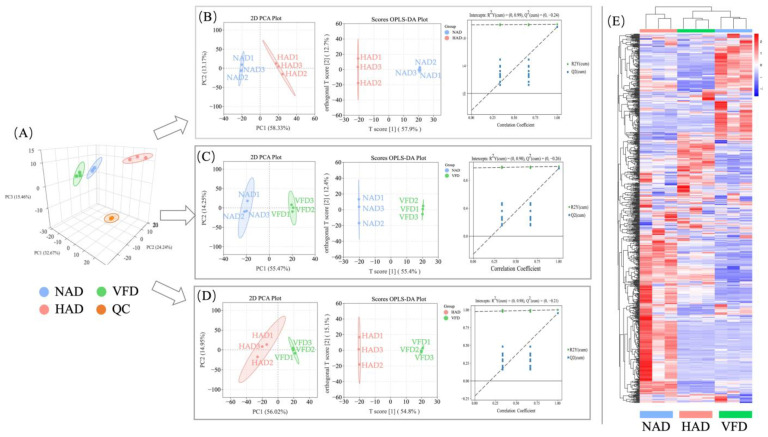
The variability of PCS differential metabolites among three different drying methods (NAD, HAD and VFD). (**A**) Three-dimensional score scatter plot of the PCA model of different PCS samples, including QC samples. (**B**–**D**) The PCA and OPLS-DA score plots and validation plots of the differential metabolites in the three pairwise comparisons (HAD vs. NAD, VFD vs. NAD and VFD vs. HAD, respectively). (**E**) Hierarchical cluster analysis of PCS metabolite contents among the three drying methods.

**Figure 3 plants-13-01594-f003:**
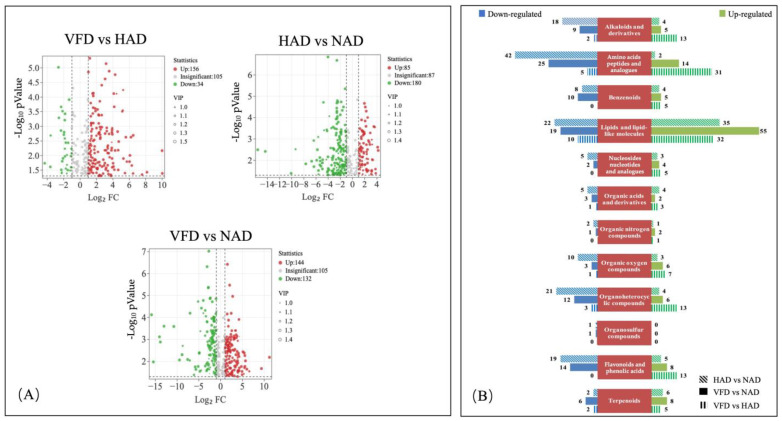
Differential metabolite analysis of the PCS of three different drying methods. (**A**) Volcano plots showing the expression levels of the differential metabolites in the three pairwise comparisons (HAD vs. NAD, VFD vs. NAD and VFD vs. HAD, respectively). Dotted lines in *x*-axis stand for log 2 (FC) = ±1 and dotted line in *y*-axis stands for *p* value = 0.05. Green dots represent down-regulated differentially expressed metabolites; red spots represent up-regulated differentially expressed metabolites; and gray spots represent non-differentially expressed metabolites. (**B**) The number of differentially expressed metabolites of three pairwise comparisons in PCS (HAD vs. NAD, VFD vs. NAD and VFD vs. HAD, respectively). Green columns represent up-regulated differentially expressed metabolites; blue columns represent down-regulated differentially expressed metabolites. The numbers located on the side of the columns represent the number of differentially expressed metabolites of each pairwise comparison of PCS. For example, the column vertical stripes of terpenoids category represent the number comparison of the terpenoids between the VFD and HAD group. The number 5 represents that in the HAD group, the number of up-regulated terpenoids is 5, which indicates that the content of 5 kinds of terpenoids in the HAD group is higher than that of the VFD group. On the contrary, the number 2 represents that in the HAD group, the number of down-regulated terpenoids is 2, indicating that the content of 2 kinds of terpenoids in the VFD group is higher than that of the HAD group.

**Figure 4 plants-13-01594-f004:**
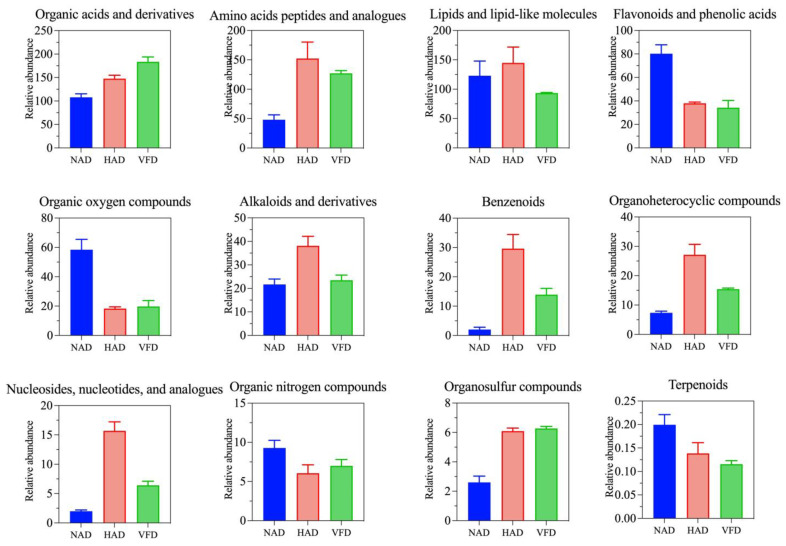
Relative abundance of the differential metabolite classes among the three drying methods.

**Figure 5 plants-13-01594-f005:**
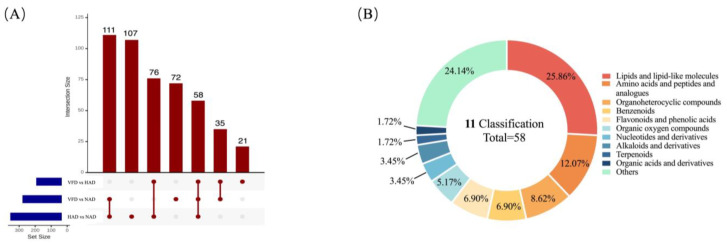
Upset diagram (**A**) displaying the overlap of differential metabolite in the three pairwise comparisons (HAD vs. NAD, VFD vs. NAD and VFD vs. HAD, respectively). And pie diagram (**B**) displaying the classification of the 58 key metabolites.

**Figure 6 plants-13-01594-f006:**
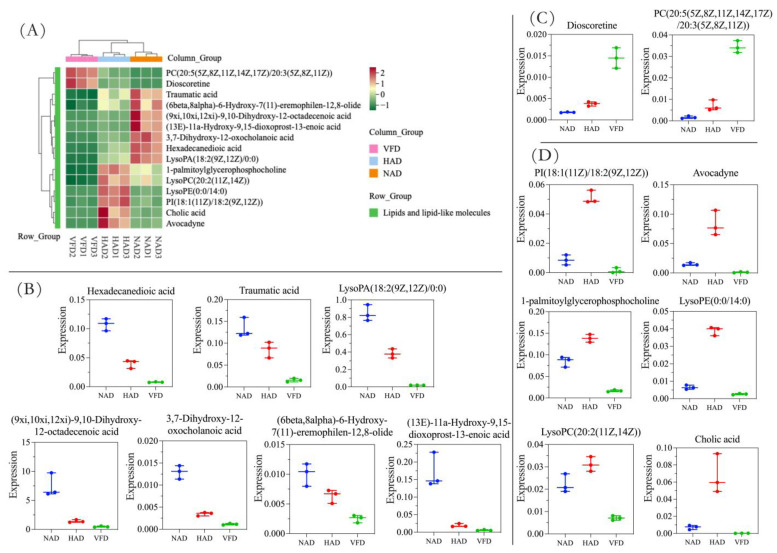
(**A**) Heatmap showing the accumulation pattern of lipids and lipid-like molecules among PCS obtained from 3 drying methods. (**B**–**D**) Boxplots showing the relative abundance of 15 differential DMs of lipids and lipid-like molecules (data are expressed as mean ± SD and *y*-axis represents the value of relative quantitative level). The lipids and lipid-like molecules with the highest expression in the NAD group are shown in figure (**B**), the lipids and lipid-like molecules with the highest expression in the VFD group are shown in figure (**C**), and the lipids and lipid-like molecules with the highest expression in the HAD group are shown in figure (**D**).

**Figure 7 plants-13-01594-f007:**
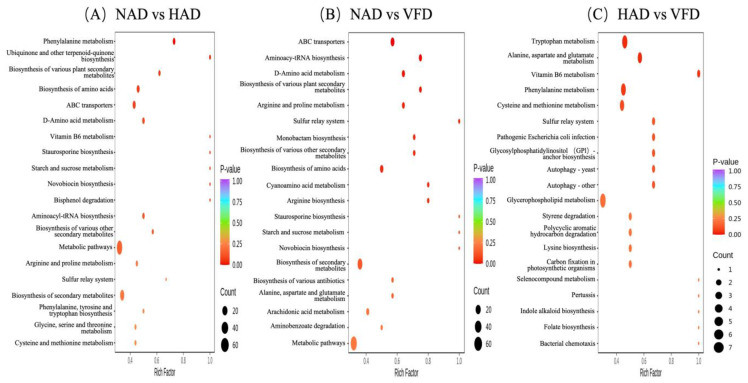
Metabolomic enrichment pathway analysis: (**A**) NAD vs. HAD, (**B**) NAD vs. VFD and (**C**) HAD vs. VFD.

## Data Availability

Data is contained within the article.

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
