# Peer review of "Exploring the Effects of Different Drying Methods on Related Differential Metabolites of Pleurotus citrinopileatus Singer Based on Untargeted Metabolomics"

_plants, 2024, doi:10.3390/plants13121594_

Round 1

Reviewer 1 Report

Comments and Suggestions for Authors

I have carefully examined the manuscript entitled “Exploring the effects of different drying methods on related differential metabolites of Pleurotus citrinopileatus Singer based on untargeted metabolomics” by Lu et al.

In this manuscript authors analyzed the difference in the metabolic profile of the edible fungus Pleurotus citrinopileatus Singer (PCS), well-known in China to have a therapeutic potential, submitted to different drying methods. They applied an untargeted metabolomic approach to measure the contents in secondary metabolites of the fungus using three drying methods, that are natural air drying (NAD), hot air drying (HAD), and vacuum freeze drying (VFD).

Their data were submitted to principal component analysis (PCA), orthogonal projections to latent structures-discriminant analysis (OPLS-DA) and hierarchical cluster analysis (HAC), for multivariate statistical analysis.

885 metabolites were detected and identified, and the results gave evidence of a variation in content of some metabolites mainly belonging to the classes of amino acids and lipids, considered to have a key role in the PCS medicinal and food applications.

I believe that this study has been properly carried out and the whole data seem to be sound, even if I need some explanations on the experimental procedure that authors should clarify.

The manuscript is well-written, and the work could be considered a useful contribution in the context of the pre-treatment of raw natural samples prior extraction of their secondary metabolites, also giving an opportunity to potential industrial applications.

Major concerns:

1. Authors must clarify how many extractions were performed. It appears, from the PCA plot, that one extraction was done for each treatment, with following three injections by LC-MS, but this is not explicitly stated in the text. To ensure statistical validity, it's crucial to have at least a triplicate extraction for each treatment. (this is a major concern)

2. Further clarification is needed regarding the metabolite identification process, particularly concerning the parameters used with the in-house database. Additionally, more detailed information is required regarding the pre-processing of LC-MS data prior to Multivariate Data Analysis. Could you specify the number of variables and observations present in the data matrix, and if you consider the peak area for each metabolite?

3. Regarding Fig 3, Authors use "downregulated" and "upregulated" for the metabolites, but these terms might not be the most suitable for describing changes in metabolite abundance. Unlike gene expression, where upregulation and downregulation refer to changes in transcription levels, metabolites don't undergo the same regulatory mechanisms. Instead, it would be more accurate to describe changes in metabolite levels as "decreased" or "increased" rather than using terms associated with gene expression.

4. Fig 4 is not mentioned in the text. Furthermore, the units on the Y-axis are not specified in this figure, leaving readers uncertain about what exactly is being measured. It's crucial for the authors to provide a clear explanation of Fig 4 in the text, including the units used on the Y-axis, to ensure the results are properly interpreted and understood.

On this basis, I consider that the above-mentioned manuscript could be suitable for publication on Plants once Authors will provide the right answers to my concerns and the following experiments/modifications required.

Comments on the Quality of English Language

The manuscript is well written, and only  minor editing of English should be required

Reviewer 2 Report

Comments and Suggestions for Authors

Translation Version

There are comments regarding the classification of compounds.

 Table S3

14 Dihydrocumambrine A – not Lipids and lipid-like molecules, – organoheterocyclic compounds.

96 - m(S,E)-Zearalenone - not Lipids and lipid-like molecules, it is a mycotoxin!

74 Acetylglycine - not Lipids and lipid-like molecules, - Amino acids and peptides and analogues

34 Sakacin P - not Lipids and lipid-like molecules - organoheterocyclic compounds

94 - Sterebin A - not Lipids and lipid-like molecules - organoheterocyclic compounds

121 - Bakkenolide C - not Lipids and lipid-like molecules, - organoheterocyclic compounds.

Nucleotides and derivatives

6 Trehalose - no, these are organic oxygen compounds.

9 Sucrose – no

Alkaloids and derivatives

1 Adenine - Nucleotides and derivatives

Organic acids and derivatives

3-Methoxytyrosine - amino acids and peptides and analogues

What explains the appearance of Fusarium fungal mycotoxins?

(S,E)-zearalenone, fumonisin B1

 It is not clear why so many compounds are given, although we are talking only about fatty acids, lipids and amino acids.

Fig. 6 is not readable.

Reviewer 3 Report

Comments and Suggestions for Authors

Mushrooms of the Pleurotaceae family are important species for human food, especially in Asian and European countries. Furthermore, the beneficial effects of these mushrooms on human health have been proved many times. Therefore, the topic of this manuscript, dealing with Pleurotus citrinopileatus constituents affected by different types of drying processes, is interesting and important.

The authors used adequate methods and an experimental design. The strength of the paper is the illustrative and detailed presentation of results in many figures and tables. Moreover, the results support the conclusions. Furthermore, the authors cited the most relevant references.

I found no serious weaknesses or mistakes in the manuscript. A minor issue: I recommend showing Figure 3 separately in several larger figures because this figure is not well visible and difficult to understand for the readers in this form. Therefore, I recommend this paper for publication.

Reviewer 4 Report

Comments and Suggestions for Authors

The manuscript by Lu et al is a descriptive paper on the compositional changes in the hydrophilic metabolome during desiccation of a mushroom species. The paper is clearly written and the results are clearly presented. I have only a few minor comments and therefore recommend that the manuscript be accepted.

Line -20-23: The sentence is difficult to understand. Please reformulate.

Line 74, Line 136: The name of P. ostreatus should be written in italics.

Line 76: please reformulate: “In He et al. (2023) research, they think that”.

Line 102, Line 136: The name of Pleurotus citrinopileatus should be written in italics.

Section 2.3: Did you perform the three drying processes several times or only once? Please include this information in this section.

Section 2.4.1: Which internal standards have been added to the extraction solution? Please specify the internal standards and their concentrations in the text.

Line 143: Are you sure you used -40oC for incubation? Was it not +40oC?

Line 153: Dimension is missing. Could this be 5mM formic acid or 0.5% formic acid?

Section 2.4.2: Define the flow rate, gradient profile and column temperature.

Line 165: Please indicate in the text if you have published your own DB.

Line 139: The name of Cinnamomum cassia should be written in italics.

Line 404, 409: typo

Reviewer 5 Report

Comments and Suggestions for Authors

The paper deals with a metabolomics study aimed at evaluating the differences in metabolites profiling obtained when drying with three different drying methods (Natural air drying, NAD, Hot air drying, HAD and Vacuum freeze drying, VFD)  Pleurotus citrinopileatus Singer (PCS), a mushroom attracting increasing interest as a raw material for medicine and food. The work performed by UHPLC-QE-MS showed 15 lipids and 7 amino acids as key discriminant metabolites in response to drying treatments out of 885 metabolites tentatively identified and 480 screened as discriminant. Despite the samples were all subject to drying methods after collections, a pathway analysis was also performed. This latter highlighted 16 possibly involved pathways, indicating the amino acid metabolism as a possible key metabolites pathway related to the PCS drying process. The paper is interesting but requires some revisions, key clarifications and possible focusing for a better outcome and improved scientific consistency.  

First of all the reader has to reach the Materials and Methods section in order to understand what kind of analytical technique (UHPLC-QE-MS ) has been used. Please report it both in the abstract and in the introduction.

Then in several point it appears that an absolute quantitative analysis was performed, nevertheless it is not clear how absolute quantitative values (if this is the case) were obtained neither (according to the captions) what the reported Y units in Fig5 B C D;  Fig S3 B C D; Fig S4 A B refer to.

Moreover also when clearly no absolute values are given, it is not clear, according to the captions, what the percentages refer to (see  Fig 3 D; Fig S2) or the bar length (see Fig 3 B) or the y units (see Fig 4).

Finally, since the samples were all subject to drying methods after collections, the reason to perform a pathway analysis is apparently related to the fact that “drying methods can exert an important effect on metabolite formation”. The Authors should better explain how this effect can arise and be related to biochemical pathway analysis being the drying methods essentially physical methods operating on a not naturally living substrate (in the case of VFD even possibly little affecting the substrate). Furthermore, on the other hand, the possible loss of volatile metabolites according to the used drying methods should be at least discussed.

A minor point, besides some typos occurring in the manuscript, Tables S2b, S2c, S3 and S4 appears mislabelled in the excel file.

Comments on the Quality of English Language

 Minor editing of English language and typos correction required

Reviewer 6 Report

Comments and Suggestions for Authors

I was tasked to review the research article “Exploring the effects of different drying methods on related differential metabolites of Pleurotus citrinopileatus Singer based on untargeted metabolomics” for Plants. The article describes a study to characterize a very interesting fungi submitted to different drying approaches (natural air, hot air, and vacuum freeze) using untargeted metabolomics based on LC-MS and chemometrics. The article is well-written, complete and propose an in-depth study of drying strategies for mushrooms of nutraceutical value. This kind of research can give a relevant added value to the associated industrial sector. I have some minor comments about this paper which I really appreciated. Generally, the authors often discuss about sensory properties without bearing in mind that this study is not suitable for olfactory studies and covers only a small number of odor-active compounds. In addition, the sample set is limited and the statistical, despite its quality, is not robust. Below some specific comments.

-        Line 86. “Metabolome” or “metabolomics”?

-        Line 183. What kind of replicates? 3 extractions of different aliquots of the same dried grinded sample or 3 full replicates including the drying procedure?

-        Fig. 2. Multivariate analyses are nice. However, did the authors evaluated fresh samples? It is important to monitor precursors evolution of the composition.

-        Fig. 4. I expect that organic oxygen compounds increase in HAD whereas I expect them to be negligible in VFD? How can the authors explain this?

-        Line 359. This is a tricky statement. There are so many compounds with a limited sensitivity in ESI ionization that can be hidden even at mild concentration. To analyze them an untargeted approach based on HS-GCxGC-MS is required. Without them the authors can study only polar compounds. An overview of these kind of methods can be found in this recent review (https://doi.org/10.3390/analytica4030026).

-        Line 395. Volatile carbonyls are usually stored in bound form so a relevant amount of them can be released due to thermal processes. This distribution was deeply studied for fermented beverages. I invite the authors to evaluate these aspects which are extensively discussed in this recent article https://doi.org/10.1021/acs.jafc.2c07083. The authors can refer to it.

Round 2

Reviewer 1 Report

Comments and Suggestions for Authors

Authors responded satisfactorily to my comments. The manuscritpt is then in a an acceptable form for publication

Comments on the Quality of English Language

Language form seems to be appropriate

Author Response

Thank you very much for your response and for recognizing this manuscript.

Reviewer 5 Report

Comments and Suggestions for Authors

The Authors have only partially answered (Comments 1) to the reviewer comments/requests.

In particular, they have completely changed the Figures composition and numbering. Therefore, the answer to the reviewer, related to the specific points of Comments 2, is unclear and it should be arranged accordingly, indicating the operated modifications and the performed clarification in the new figures.

“Then in several point it appears that an absolute quantitative analysis was performed, nevertheless it is not clear how absolute quantitative values (if this is the case) were obtained neither (according to the captions) what the reported Y units in Fig5 B C D;  Fig S3 B C D; Fig S4 A B refer to.

Moreover also when clearly no absolute values are given, it is not clear, according to the captions, what the percentages refer to (see  Fig 3 D; Fig S2) or the bar length (see Fig 3 B) or the y units (see Fig 4).”

Moreover, the answer to Comments 3 should clarify the key issue:

 “The Authors should better explain how this effect can arise and be related to biochemical pathway analysis being the drying methods essentially physical methods operating on a not naturally living substrate (in the case of VFD even possibly little affecting the substrate).”

Furthermore, the other suggestion reported in Comments 3 has not be taken into account in the revised manuscript.

"the possible loss of volatile metabolites according to the used drying methods should be at least discussed."

Comments on the Quality of English Language

Minor editing of English language required

Round 3

Reviewer 5 Report

Comments and Suggestions for Authors

The Authors have essentially answered to the reviewer comments/requests

Comments on the Quality of English Language

Minor editing of English language required which could be performed at the  proof preparation stage

Author Response

(The authors gave the same response as above.)
